**Retrieval of dominant methane (CH₄) emission sources, the first high resolution**
**(1-2m) dataset of storage tanks of China in 2000-2021**
Fang Chen[1, 2, 3, †], Lei Wang [1, 2, 4, †], Yu Wang[5, 6, *], Haiying Zhang[1, 2], Ning Wang[7],
Pengfei Ma[5, 6], Bo Yu [1, 2, 4, *]
[1]International Research Center of Big Data for Sustainable Development Goals, Beijing,
100094, China
[2]Key Laboratory of Digital Earth Science, Aerospace Information Research Institute,
Chinese Academy of Sciences, Beijing, 100094, China
[3] University of Chinese Academy of Sciences, Beijing, 100049, China
[4] School of Computer Science and Information Security, Guilin University of Electronic
Technology, Guilin, 541004, China
[5] State Environmental Protection Key Laboratory of Satellite Remote Sensing, Beijing
100094, China;
[6] Satellite Application Center for Ecology and Environment, Ministry of Ecology and
Environment, Beijing 100094, China;
[7]College of Urban and Environmental Sciences, Peking University, Beijing, 100871,
China
† These authors contributed equally to this work and should be considered as co-first
authors
*Corresponding author: Yu Wang (chenfang_group@163.com), Bo Yu
(yubo@radi.ac.cn)
**Abstract.** Methane (CH₄) is a significant greenhouse gas in exacerbating climate
change. Approximately 25% of CH₄ is emitted from storage tanks. It is crucial to
spatially explore the CH₄ emission patterns from storage tanks for efficient strategy
proposals to mitigate climate change. However, due to the lack of publicly accessible
storage tank locations and distributions, it is difficult to ascertain the CH₄ emission
spatial pattern over a large-scale area. To address this problem, we generated a storage
tank dataset (STD) by implementing a deep learning model with manual refinement
based on high spatial resolution images (1-2m) from the GaoFen-1, GaoFen-2, GaoFen-
6, and Ziyuan-3 satellites over cities in China with officially reported numerous storage
tanks in 2021. STD is the first storage tank dataset over 92 typical cities in China. The
dataset can be accessed at https://zenodo.org/records/10514151 (Chen et al., 2024). It
provides a detailed georeferenced inventory of 14,461 storage tanks, wherein each
storage tank is validated and assigned the construction year (2000-2021) by visual
interpretation referring to the collected high spatial resolution images, historical high
spatial resolution images of Google Earth, and field survey. The inventory comprises
storage tanks having various distribution patterns in different cities. Spatial consistency
analysis with CH₄ emission product shows good agreement with storage tank
distributions. The intensive construction of storage tanks significantly induces CH₄
emissions from 2005 to 2020, underscoring the need for more robust measures to curb



CH$_4$ release and aid in climate change mitigation efforts. Our proposed dataset STD
will foster the accurate estimation of CH$_4$ released from storage tanks for CH$_4$ control
and reduction and ensure more efficient treatment strategies are proposed to better
understand the impact of storage tanks on the environment, ecology, and human
settlements.

**1. Introduction**


The Industrial Revolution witnessed a continuous increase in greenhouse gases,
resulting in global climate warming (Zhang et al., 2021). Methane (CH$_4$) is the second
dominant anthropogenic greenhouse gas to global climate warming with a contribution
of 20% (Kirschke et al., 2013) after carbon dioxide (CO$_2$). Meanwhile, CH$_4$ is more
effective in trapping heat, with 85 times more climate warming potency than CO$_2$
(Stocker, 2014). The atmospheric lifetime of CH$_4$ is approximately 10 years, which is
shorter than most other greenhouse gases; thus, reducing CH$_4$ emissions is more cost-
effective in lowering the climate warming potential impact (Lin et al., 2021; Montzka
et al., 2011). CH$_4$ is emitted mainly from energy-related activities and petrochemical
processes (Ding et al., 2017; Fan et al., 2023). Storage tanks,    defined as large
containers of crude oil or other petroleum, and industrial materials, such as alcohols,
gases, or liquids, are among the most significant sources of emitting CH$_4$ (Im et al.,
2022; Johnson et al., 2022). Without an adequate control or management strategy, large
amounts of CH$_4$ will escape into the atmosphere (Im et al., 2022). From a greenhouse
gas control standpoint, it is of great interest to examine the distribution patterns of the
storage tanks. With a detailed and comprehensive storage tank inventory, we can
effectively estimate the spatial pattern of CH$_4$ emissions and reduce the risk of CH$_4$
emission by installing recovery units (Johnson et al., 2022) to promote sustainable
development goals. However, it is challenging to access detailed distribution records
for storage tanks from the public records in China.
Given the advances in remotely sensed technology (Chen et al., 2023; Yu et al.,
2023a; Yu et al., 2023b), the ready availability of high spatial resolution remote sensing
images via the GaoFen series satellites and the Ziyuan-3 satellite provides means to
extract remote sensing data for large-scale storage tanks. Numerous studies on the use
of automatic methods to extract storage tanks from high spatial resolution remote
sensing images have been performed (Fan et al., 2023; Wu et al., 2022; Yu et al., 2021),
including the Hough transform (Yuen et al., 1990), image saliency enhancement (Zhang
and Liu, 2019), support vector machines (Xia et al., 2018), and Res2-Unet+ deep
convolution networks (Yu et al., 2021). The focus of the works above is primarily
spatially limited, and the images collected for extraction are mostly pre-subtracted from
regions known to contain storage tanks. The transferability and the practical
applicability of the proposed methods remain to be clarified. To our knowledge, there
are limited publicly available datasets on storage tanks. Northeast Petroleum
University–Oil Well Object Detection Version 1.0 (NEPU–OWOD V1.0) covers 1,192
oil storage tanks within Daqing City (Wang et al., 2021). This dataset covers the
boundary boxes for each storage tank but lacks details on the storage tank inventory.
Another two datasets, the Oil and Gas Tank Dataset (Rabbi et al., 2020) and the Oil
Storage Tank Dataset (Heyer, 2019) acquired via the Kaggle platform, have been
released without georeferenced information and lack detail regarding the contour
shapes. The datasets are generally proposed to improve the performance of algorithms
in storage tank extraction. Currently, most studies are concentrated on algorithm
development for storage tank extraction rather than exploring the spatial distribution of
storage tanks in large-scale areas and the impact of storage tank construction on $CH_4$
emission in different areas over the years. The spatial distributions of storage tanks in
China have not yet been investigated and recorded. The lack of storage tank datasets
makes it impossible to estimate the impact of anthropogenic energy-related activities
on $CH_4$ emission and air pollution.
To foster the control and reduction of $CH_4$ emissions to mitigate climate change
and provide researchers with free access to detailed and georeferenced storage tank
inventory to monitor the corresponding potential impact on the atmosphere and
residential environment over typical cities in China, we compiled a storage tank
inventory based on high spatial resolution images of the GaoFen-1, GaoFen-2, GaoFen-
6, and Ziyuan-3 satellites for cities with intensive storage tanks over China. The cities
are listed by the Ministry of Ecology and Environment of China with intensive storage
tanks and prominent fugitive emissions, inadequate monitoring and control of treatment
measures (Wang et al., 2022). There are 92 cities in total, mainly located in mid-eastern
China. Given that large storage tanks may emit significant levels of $CH_4$, storage tanks
of size $\geq 500$ m$^2$ were selected as the main target to control the reduction of $CH_4$ in the
proposed inventory. To this end, we generated a complete inventory of storage tanks of
size $\geq 500$ m$^2$ for the 92 cities in China with intensive storage tanks, which were subject
to the implementation of $CH_4$ reduction measures.
In this study, firstly, we collected high spatial resolution images to cover the entire
study area. We pre-processed them to synchronize the pixel intensities of ground objects
in different images from different imaging sensors and study areas. Secondly, we
proposed a semantic segmentation framework to construct the storage tank extraction
model based on the training samples of Ningbo, Tangshan, and Dongying cities. Thirdly,
the constructed model is applied to extract storage tanks in all the other cities to generate
extraction results. Fourthly, the extracted storage tank result images are converted to
vectors, revised and assigned the corresponding construction year by visual
interpretation with reference to the historical high spatial resolution images of Google
Earth, high spatial resolution images collected, and field survey. Fifthly, we explored
the spatial distribution pattern of storage tanks in typical cities in China. Sixthly, we
further explored the consistency of storage tank spatial patterns and $CH_4$ emission in
the atmosphere and the impact of storage tank construction on time-series $CH_4$ emission
change from 2005 to 2020. Finally, the uncertainties, limitations, and implications of
our proposed STD dataset are discussed for studying climate change and air pollution.
This new database represents the first inventory to provide a detailed distribution of the
locations, boundaries of the storage tanks, and the corresponding construction year of
each storage tank. The inventory documents the spatial and temporal distribution of
storage tanks of different sizes, and it is hoped that this work will facilitate the
development of environment-friendly regulatory proposals for more effective $CH_4$



emission control and energy resource management.

## 2. Related works in mapping storage tanks

Storage tank extraction from high spatial resolution images has been of interest for
many years for its significant role in storage and greenhouse gas emission. Generally,
the methods for extracting storage tanks are grouped into three categories. Circle
detection by Hough transformation (O'duda, 1972) and template matching (Hou et al.,
2019); machine learning model construction by morphological, spectral, and textual
feature engineering (Xia et al., 2018); deep learning model construction by continuous
convolution operations (Fan et al., 2023). Deep learning methods have been extensively
used to map storage tanks due to their strong feature learning capability and higher
model transferability.
Semantic segmentation is a widely employed deep learning framework in object
extraction by assigning each pixel a semantic label in the image (Chen et al., 2022; Yu
et al., 2022b). Fully convolution network (FCN) (Long et al., 2015) is a basic
framework of semantic segmentation with three components: backbone feature learning,
convolution feature learning with skip architecture, and up-sampling layer to resample
the learned feature map to the same size of the input image. Based on FCN, numerous
frameworks have been inspired, such as SegNet (Badrinarayanan et al., 2017), PSPNet
(Zhao et al., 2017), Unet (Ronneberger et al., 2015), DeepLabv2 (Chen et al., 2017b),
and DeepLabv3 (Chen et al., 2017a). Unet has a widespread use for its easy
implementation and high efficiency. The proposal of Res2-Unet+ framework for
storage tank extraction (Yu et al., 2021; Zalpour et al., 2020) integrates Res2Net module
(Gao et al., 2019) to Unet. Res2Net module is proposed to learn multi-scale features by
learning at a more granular level. It has shown strong applicability in extracting storage
tanks from images of different imaging sensors (Yu et al., 2022a). However, many
storage tank pixels are still omitted due to their similar spectral characteristics with
neighboring ground objects. To resist the shortage, we have proposed a new semantic
segmentation framework based on Res2-Unet+ and enlarged the variability of storage
tank training samples to build a more robust and accurate extraction model.

## 3. Data sources

### 3.1 Study area

The study area covers 92 typical cities (as shown in Figure 1) with intensive
storage tanks over China, assigned by the Ministry of Ecology and Environment of
China (Wang et al., 2022). The typical cities lack detailed monitoring and control of
prominent fugitive emissions, whose effective measurements in $CH_4$ reduction
emission are urgently demanding and requiring. The 92 cities tended to be located in
mid-eastern China. Many of the cities are located near or next to the boundary of
mainland China. Synthesized with a digital elevation model (DEM) from the product
of the Shuttle Radar Topography Mission (SRTM) (Yang et al., 2011), we can recognize
that most cities are plains. As is acknowledged, plains are densely populated. The large
population numbers will bring more frequent human activities, triggering more
pollutant and greenhouse gas emissions. The lack of efficient measurements in $CH_4$
reduction will result in a more direct impact on the populations in the residential area.
Therefore, exploring the spatial distribution pattern of storage tanks relative to $CH_4$
emission is significant to seek more effective solutions for $CH_4$ reduction.

*Figure 1. Study area demonstration with digital elevation from the Shuttle Radar*
*Topography Mission (SRTM) product.*

**3.2 High spatial resolution images**
The high spatial resolution images used for extracting storage tanks in the 92 cities
were collected from four satellites: the GaoFen-1, GaoFen-2, GaoFen-6, and Ziyuan-3
satellites in 2021. The images are collected between June and August with the least
cloud coverage (<10%) from the four satellites, when different ground objects have
more pronounced spectral differences, which makes it easier to distinguish storage
tanks from background objects. As listed in Table 1, the images for the GaoFen-1,
GaoFen-6, and Ziyuan-3 satellites have a spatial resolution of 2 m, and those for the
GaoFen-2 have a spatial resolution of 1 m after fusion of the multispectral image and
the panchromatic image. Referring to Table 1, we can recognize that 4,403 images were
collected. The places covered with multiple images are manually screened to one image
with the best imaging quality and least cloud proportion. Based on the screened high
spatial resolution images, multiple image pre-processing steps are performed to

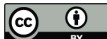

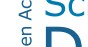
synchronize the ground objects in different images of different sensors for different
study areas, comprising atmospheric correction, radiation correction, geometric
precision correction, image fusion, image projection, uniform color processing, and
image mosaicking.

*Table 1. Imaging characteristics of each high spatial resolution satellite and the*
*number of collected images of different satellites covering 92 typical cities in China*
*between June and August 2021. The notation Pan is short for Panchromatic band,*
*and Multi represents multi-spectral band*

|  | **GaoFen-1** | **GaoFen-2** | **GaoFen-6** | **Ziyuan-3** | **Total** |
|---|---|---|---|---|---|
| **Spatial resolution** | 2m(Pan)/ 8m(Multi) | 1m(Pan)/ 4m(Multi) | 2m(Pan)/ 8m(Multi) | 2m(Pan)/ 6m(Multi) |  |
| **Multi-spectral Band** | Red/Green/ Blue/Near-Infrared | Red/Green/ Blue/Near-Infrared | Red/Green/ Blue/Near-Infrared | Red/Green/ Blue/Near-Infrared |  |
| **Number** | 1,289 | 1,330 | 139 | 1,645 | 4,403 |


**3.3 Land use land cover product**

Given that storage tanks are constructed mainly in residential areas, a 10 m land
use land cover (LULC) product of the Esri Land Cover in 2021 (Karra et al., 2021) is
used for subtracting the study area to minimize the impact of complex background
objects in the high spatial resolution images following the workflow as shown in Figure
2. The land use product of the Esri Land Cover is generated based on the Sentinel-2
images from the European Space Agency (ESA) with an overall accuracy of 75%
(Venter et al., 2022), which has been updated every year since 2017. It comprises nine
ground object categories: water, trees, flooded vegetation, bare ground, crops, snow/ice,
clouds, rangeland, and built area. Since storage tanks are mostly constructed in urban
areas, the categories of built area and bare ground are recognized as potential areas for
constructing storage tanks. Consequently, the corresponding ground object category
products of built area and bare ground are subtracted from the LULC product 2021 and
used to mask the high spatial resolution images of the 92 cities, as demonstrated in
Figure 2. Pixels locating outside the mask area in the high spatial resolution images,
whose intensities are assigned zero. The masked high spatial resolution images of the
92 cities are further used for storage tank extraction.



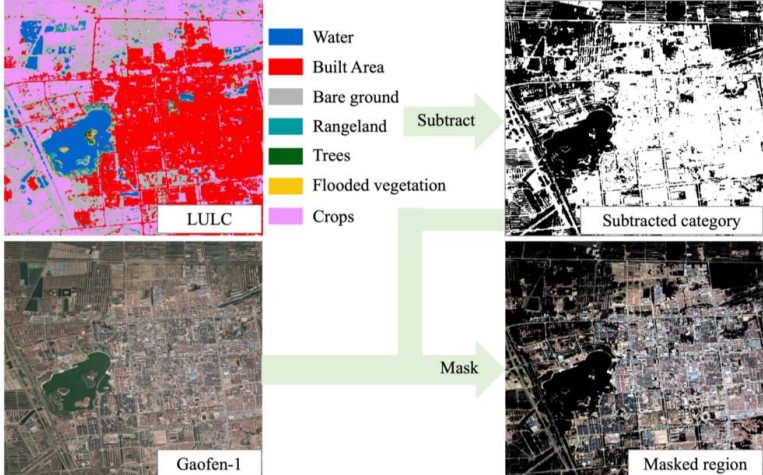

*Figure 2. Subtraction of potential area with storage tanks from high spatial resolution*
*images.*

### 3.4 CH₄ product image

As storage tanks are a dominant source of $CH_4$ emission, we have collected $CH_4$
emission products to explore the spatial consistency of $CH_4$ with the density of storage
tanks and the impact of storage tank construction over time on $CH_4$ emission. There
have been many $CH_4$ emission product images proposed, including the Community
Emission Data System (CEDS) (Hoesly et al., 2018), the product from Peking
University (Peng et al., 2016), the Emissions Database for Global Atmospheric
Research (EDGAR) (Crippa et al., 2019), the Regional Emission Inventory in Asia
(REAS) (Kurokawa et al., 2013), and Greenhouse Gas and Air Pollution Interactions
and Synergies (GAINS and ECLIPSE) (Amann et al., 2011). Since our collected high
spatial resolution remote sensing images were taken in the year 2021, the spatial
consistency and the impact of storage tank construction on $CH_4$ emission are explored
using the $CH_4$ emission product of GAINS, which offers a comprehensive series of data
accessible to the public (Lin et al., 2021). The dataset of GAINS was selected over the
other four products because the four products lacked continuous updates with limited
temporal coverage until 2015.
We adopted the estimated $CH_4$ emission from energetic activities product of the
ECLIPSE V6b Baseline scenario from GAINS. It is a global annual product with a
spatial resolution of 0.5° and a temporal coverage of 1990-2050 at an interval of 5 years.
For the estimated $CH_4$ emission from GAINS in the years 1990-2018, the product is
generated from statistics of the International Energy Agency (IEA), and the years 2019-
2050 are from the outlook of the IEA World Energy Outlook (Lane, 2018). To
synchronize with the temporal scope of storage tank construction from 2000 to 2021,
the $CH_4$ emission products of 2005, 2010, 2015, and 2020 are collected.
As demonstrated in Figure 3, the emission of $CH_4$ in 2020 varies remarkably in
different areas. There are many clusters of $CH_4$ emission in the study area, with the

highest of 5,160.62 Tg $CH_4$ $yr^{-1}$. $CH_4$ in the atmosphere of cities located in southeastern China is generally higher than that of cities in northwestern China in 2020.

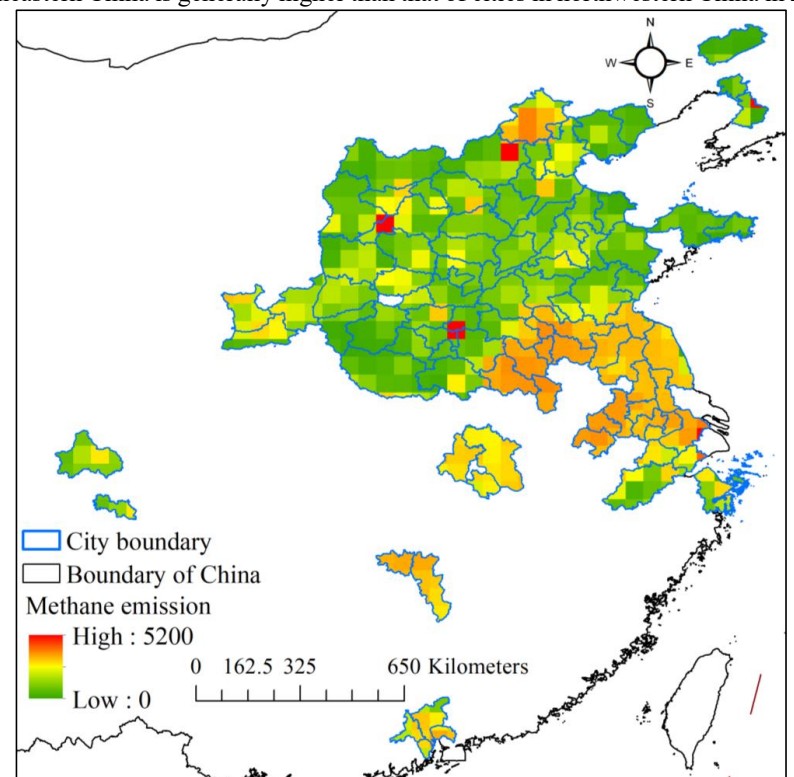

*Figure 3. Demonstration of CH$_4$ distribution from energetic activities over the study area in the year 2020.*

## 4. Methodology

As depicted in Figure 4, the workflow of generating a storage tank dataset consists of three sections: harmonizing the pixel intensities of different ground objects across high spatial resolution images captured by different sensors in different study areas; producing a storage tank dataset by constructing a storage tank extraction model based on the harmonized high spatial resolution images; assigning the construction year of each storage tank by multiple experts through visual interpretation referring to the historical high spatial resolution images, high spatial resolution images collected, and filed survey from Google Earth.



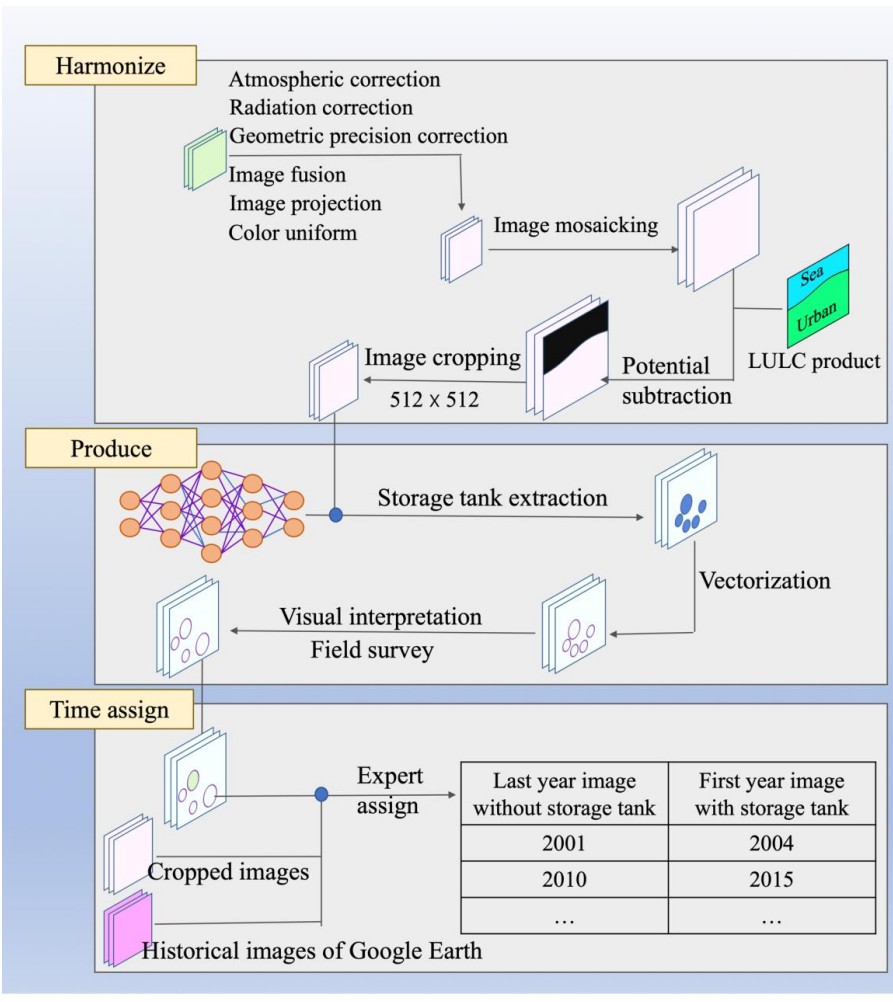

Figure 4. Flow chart for storage tank inventory production.

**4.1 Image harmonizing**

Pixel intensities for ground objects are standardized to ensure consistency across the high spatial resolution images collected. This harmonization process mitigates the effects of atmospheric variations and discrepancies between imaging sensors captured at different times. The standardization includes atmospheric correction, radiometric calibration, geometric alignment, image fusion, reprojection, and color normalization. In terms of atmospheric correction, the widely used radiation transfer model of the second simulation of the satellite signal in the solar spectrum (6S) (Vermote et al., 1997) is adopted to determine the atmospheric correction coefficient and eliminate the absorption and scattering impact of atmospheric molecules and aerosols for all the collected high spatial resolution images. The strategy of local histogram matching



(Shen, 2007) is used to correct radiation differences of the same ground object category
in different high spatial resolution images. To improve the geometric precision of the
high spatial resolution images collected, we automatically generated 1000 ground
control points by a widely used key point detector of scale-invariant feature transform
(SIFT) for each city. We calculated the parameters for affine transformation with
reference to the world imagery of Environmental Systems Research Institute (ESRI)
(Hou et al., 2021). Pixel-wise image fusion is conducted on images collected from each
high spatial resolution satellite since they consist of multispectral images with a coarser
spatial resolution than the panchromatic image, as demonstrated in Table 1. To optimize
the utilization of the gathered images, we leveraged the wavelet transform (Sahu and
Sahu, 2014) for the automatic fusion of multispectral and panchromatic images. To
address discrepancies in the projections of the varied high-resolution images we
collected, we standardized all the images to the Universal Transverse Mercator (UTM)
projection using bilinear interpolation for consistency. To maintain visual consistency
across images from different sensors or regions, it is crucial to standardize the color
representation of identical ground objects. In this study, we implemented a nonlinear
stretching technique to modify pixel intensity distribution. This was accomplished by
constructing a color look-up table (Majumder et al., 2000) to ensure uniformity in
spectral intensities across the various images.
The harmonized high spatial resolution images were further mosaicked to large
image patches to integrate overlapping areas from adjacent high-resolution images,
ensuring comprehensive coverage and continuity of the observed regions. Referring to
the LULC product of the Esri Land Cover product in 2021, the mosaicked image
patches were subtracted with the ground object category of built area and bare ground,
identified as potential areas with storage tank constructions. Finally, for storage tank
extraction, the subtracted images were cropped to a size of 512×512 pixels, a size
compatible with the computational limits of our GPU hardware.

### 4.2 Production of storage tank dataset

#### 4.2.1 Proposed framework for storage tank extraction

Stemming from the recently developed semantic segmentation framework for
storage tank extraction, Res2-Unet+ by Yu et al. (Yu et al., 2021), we proposed a new
network structure Res2-UnetA to build storage tank extraction model. As shown in
Figure 5A, our proposed framework integrates the Res2Net module (Figure 5B) and
channel-spatial attention module (Figure 5C) to enhance the significant features for
multi-scale storage tank extraction. During the process of feature map down-scaling,
the Res2Net module can learn the multi-scale features from multiple sub-networks and
concatenate the multi-scale features to enlarge the visual perception capability. In the
stage of feature map up-sampling, our proposed channel-spatial attention module
adopted after each feature map concatenation operation can increase the feature
learning efficiency and enlarge the feature learning scale by synthesizing channel-wise
and spatial attention feature learning modules. Detailed calculation of channel-wise and
spatial attention modules can be referred to Equations (1)-(7). Spatial average pooling
(sa) and spatial maximum pooling (sm) operations are calculated as the average value
and maximum value of input feature map $f$, as described in Equations (1)-(2).



Correspondingly, the channel-wise average (ca) and maximum pooling (cm) operations
are the average feature values of all the channels and the maximum feature values of
all the channels in Equations (3)-(4). The output feature map of the spatial attention
module (SA) and channel attention module (CA) are calculated according to Equations
(5)-(6), respectively, and the synthesis of the feature maps from the channel and spatial
attention modules is organized by multiplication, as illustrated in Equation (7). Through
multi-scale feature enhancement by our proposed Res2-UnetA framework, it can learn
the multi-scale storage tank features hierarchically and comprehensively from the high
spatial resolution images of the different imaging sensors.

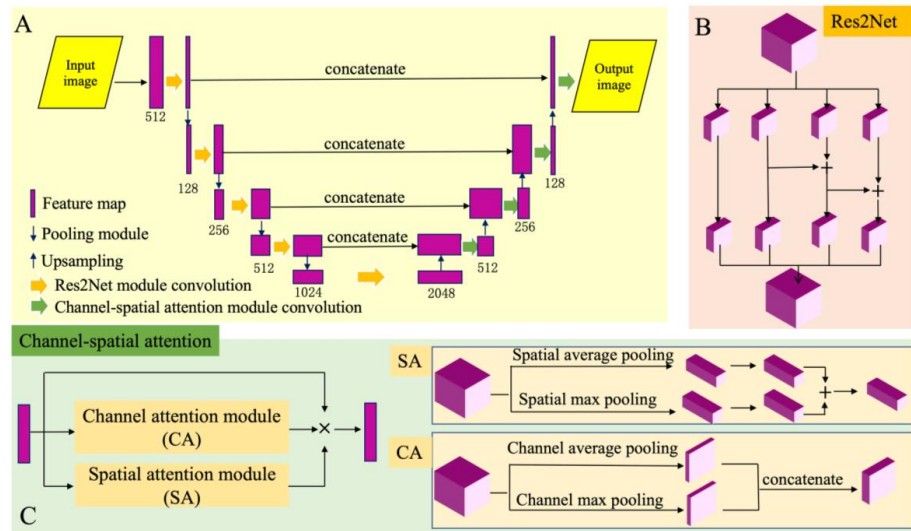

*Figure 5. Network structure of our proposed Res2-UnetA: (A) network general*
*demonstration; (B) structure of Res2Net module; (C) structure of channel-spatial*
*attention module.*

$$sa_f = \frac{\sum_{i=0}^{m}\sum_{j=0}^{n} f_{i,j}}{m \times n} \tag{1}$$

$$sm_f = \max\left(f_{i=0,\cdots,m,\,j=0,\cdots,n}\right) \tag{2}$$

$$ca_f = \frac{\sum_{c=0}^{h} f_{c=k}}{h} \tag{3}$$

$$cm_f = \max\left(f_{c=0,\cdots,h}\right) \tag{4}$$

$$SA(f) = conv\left(conv\left(sa_f\right) + conv\left(sm_f\right)\right) \tag{5}$$

$$CA(f) = conv\left(concatenate\left(ca_f, cm_f\right)\right) \tag{6}$$

$$CSA(f) = f \times CA(f) \times SA(f) \tag{7}$$

**4.2.2 Storage tank model construction and dataset generation**



Based on our proposed framework Res2-UnetA, the pre-processed high spatial
resolution images for the cities of Ningbo, Tangshan, and Dongying are used to train
the storage tank extraction model. Ningbo, Tangshan, and Dongying are three typical
cities in China with large densities of storage tanks so that they can provide large
quantities of training samples with high spectral and textual feature variety in different
sizes. The storage tanks for the training dataset are interpreted visually by three experts
in a relative field referring to the collected high spatial resolution images. The model is
finetuned based on the optimized model from Res2-Unet+ by Yu et al. (Yu et al., 2021)
with a learning rate of 0.01. It converges to the optimum at the iteration of 69.

With the optimized model, the storage tanks for the remaining cities are extracted
accordingly and vectorized to the shapefile. While the enhanced model for extracting
storage tanks generally performs well, it's not infallible. Some tanks are inadvertently
missed, and other objects with similar spectral or textural characteristics are
occasionally mistakenly identified as tanks. Therefore, the vectorized shapefile is
further refined manually by visual interpretation with referral to the high spatial
resolution images. Due to the inconsistent spectral intensities for the storage tanks in
the images, triggered by shadows and different viewing angles, the vectorized storage
tanks in the inventory take different shapes. To synchronize the storage tanks in the
inventory taking on a round shape, we re-construct a circle for each extracted storage
tank according to the radius calculated in the inventory, and the inventory is updated
with the re-constructed circle. To facilitate the dating of each storage tank's construction
year, the reconstructed circle for each extracted storage tank has been manually
validated by six experienced experts through visual interpretation based on our
collected high spatial resolution images and field survey.
**4.3 Construction year assignment**

In the STD dataset we developed, a team of six experts determines the construction
year for each storage tank by conducting visual assessments using high-resolution
historical images available on Google Earth, with the cutoff date for this process being
January 1$^{st}$, 2024. The intermittent availability of historical high-resolution images on
Google Earth poses a challenge in determining the precise construction years for many
storage tanks, especially when images from successive years are missing. We
documented the most recent year when a storage tank was absent (last year image
without storage tank) and the earliest year when it was first observed (first year image
with storage tank) in the historical imagery, as illustrated in Figure 4. The actual
construction year lies within this timeframe. For analysis simplicity, we've designated
each tank's initial observed year as the construction year.

Since the high-resolution images used to compile the storage tank dataset were
captured in 2021, it is presumed that all tanks were constructed no later than this year.
However, due to the absence of updated high-resolution imagery on Google Earth, 488
tanks remain undetected in the historical records. For these, the year of construction has
been inferred as 2021, following thorough visual confirmation using the high-resolution
images we have acquired. The considerable lapses in historical high-resolution imagery
on Google Earth necessitate assigning a provisional construction year 2021 to 630
storage tanks. The year of 2021 marks the earliest documented evidence of these tanks'



existence in the high-resolution images we collected, beyond which no prior images are
available.

**5. Results**
**5.1 Spatial distribution of storage tanks**
Following the workflow in Figure 4, the storage tanks in the 92 typical cities of
China are extracted based on the high spatial resolution images using the trained
semantic segmentation model. Given that large capacity storage tanks are known to
release significant levels of $CH_4$, resulting in climate warming, the proposed inventory
focuses on storage tanks with an area of 500 $m^2$. 14,461 storage tanks are extracted
from the 92 cities with areas ranging from 500 $m^2$ to 18,583.15 $m^2$. As shown in Figure
6, the storage tanks are distributed unevenly in different cities and reflect different sizes
and spatial distribution patterns. To explore the different distribution patterns, the
storage tanks are categorized into three groups according to the area: 500-1,000 $m^2$,
1,000-10,000 $m^2$, and ≥10,000 $m^2$. The accumulated number of storage tanks of
different sizes for all the cities is compiled as shown in Figure 7. It may be seen that
storage tanks of 500-1000 $m^2$ are more than those of larger sizes. The relatively smaller
storage tanks are more widely used in industry. Due to the high cost of construction,
considering all the cities, the maximum number of large storage tanks of size ≥10,000
$m^2$ is found to be seven for the city of Tangshan. Notably, there are few cities with
storage tanks of 10,000 $m^2$ in size.





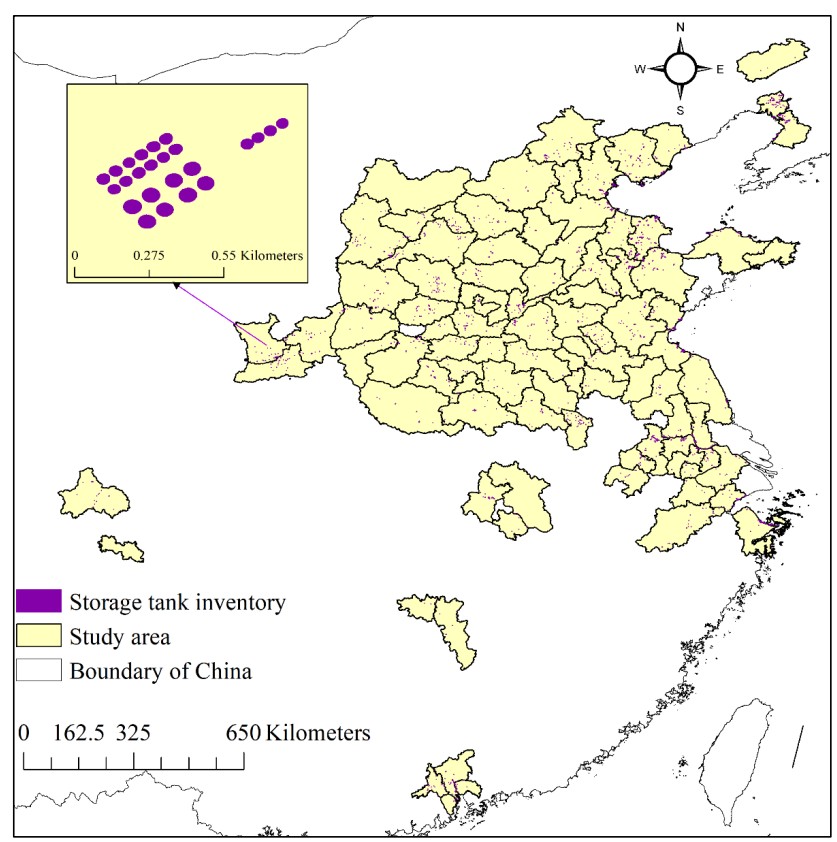


*Figure 6. Inventory for storage tanks of the 92 typical cities.*

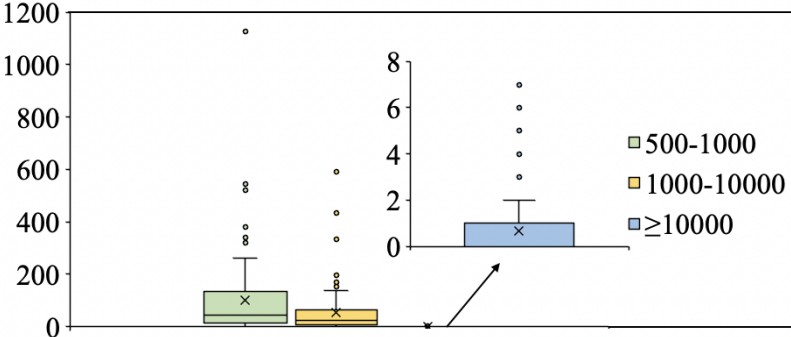


*Figure 7. Box plot of storage tank distribution for the different size categories ($m^2$) for*
*the 92 cities.*

About the 92 cities examined, 38 cities have storage tanks with an accumulated

$\geq 100$, as shown in Figure 8A. Dongying has the largest accumulated number of 1719,



about twice that of Ningbo, the second highest ranked city with 981 storage tanks.
Weifang and Panjin are next in rank with storage tanks more than 500. The number of
storage tanks of size 500-1000 m² is greater than that for 1,000-10,000 m² and ≥10,000
m² for most cities. This finding indicates the widespread use of smaller storage tanks in
different industries. Furthermore, there are 36 cities with an accumulated number of <
50 (Figure 8B). Among the 36 cities, Hebi is the only city with four storage tanks of
≥10,000 m² in size. The other cities, except Tangshan, do not have that large storage
tanks. No storage tanks of size ≥500 m2 are observed for the cities of Taian, Weihai,
and Zigong.

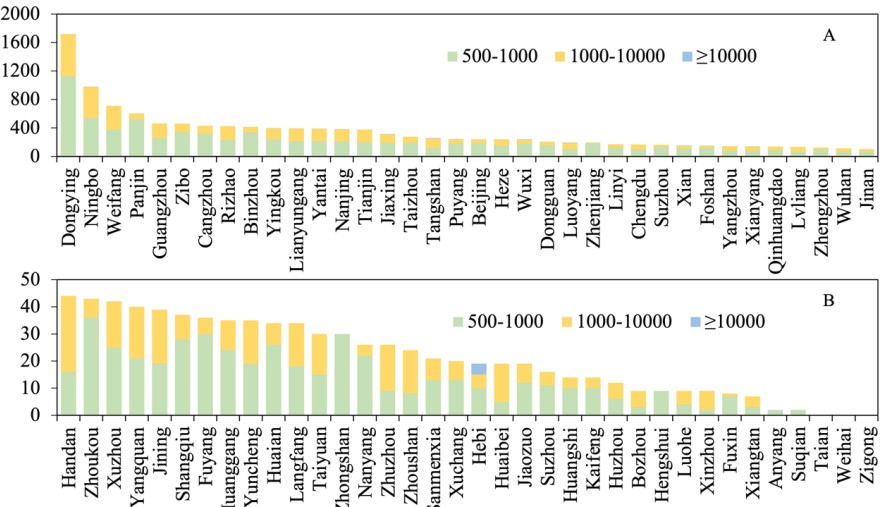

*Figure 8. Number of storage tanks of different size categories in the various cities: (A)*
*cities with an accumulated storage tank number ≥100; (B) cities with accumulated*
*storage tank number of <50.*

## 5.2 Spatial consistency with CH₄ emission

To explore whether the distribution patterns of storage tanks influence CH₄
emissions significantly, we explored the spatial consistency between estimated CH₄
from energy emission products and the density of storage tanks in our proposed dataset
STD over the study area. Given the coarser spatial resolution of the CH₄ emission
product at 0.5°, which is less detailed than that of the high spatial resolution images
used for generating our storage tank dataset, we've calculated storage tank density to
align with each pixel grid of the CH₄ data. The density is defined by the total storage
tank area ratio within each corresponding 3025 km² pixel grid area (55km × 55km),
where 55 km is an approximation of 0.5° latitude or longitude at the equator.
The storage tank density is calculated for each grid pixel of the CH₄ emission
product and is demonstrated in Figure 9. We can recognize that large-scale areas with
high CH₄ emission in the atmosphere generally cluster large densities of storage tanks
(clustered cases of A, B, C, and D). The sparsely distributed storage tanks with high
density are mostly accompanied by a higher CH₄ emission than the neighborhood (as



shown in cases of E). There are also some cities with a high density of storage tanks
and low CH$_4$ emission estimation, especially at the border of mainland China, as in the
cases of F. That could be attributed to the coastal air currents, which will likely disperse
CH$_4$ emissions more effectively. It also needs to be pointed out that for the cities marked
as G in Figure 9, the estimated CH$_4$ emission is relatively high, but the density of storage
tanks is low. One possible reason is the unrestrained leakage of CH$_4$ from the storage
tanks, highlighting the urgent need for effective control measures. Alternatively, other
high-energy activities within these regions might be significant CH$_4$ contributors,
suggesting a need for comprehensive investigation into broader mitigation strategies.

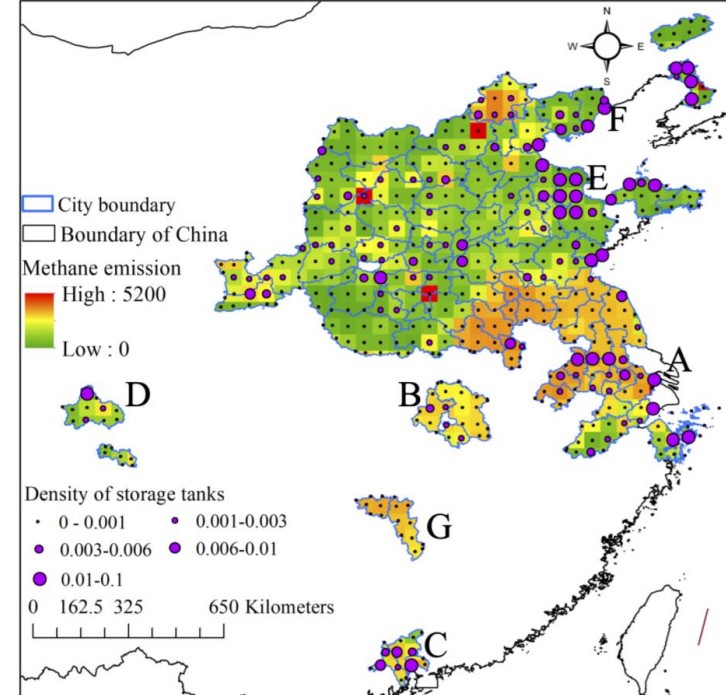

*Figure 9. Spatial distribution pattern of different densities of storage tank area with*
*different CH$_4$ emissions in the atmosphere.*

To objectively explore the spatial consistency of storage tank distribution and CH$_4$
emission from energetic activities, we randomly selected 4000 storage tank pixels and
4000 background object pixels to evaluate the significance of the impact of storage
tanks on CH$_4$ emission. Referring to Figure 3, the value of CH$_4$ emission varies by a
large margin between 0.000055 and 5160.32 Tg CH$_4$yr$^{-1}$. The large value gap of CH$_4$
emission will cause bias in the differential significance test. We generated the quantity
distribution of pixels with different CH$_4$ emission value gaps (as shown in Figure 10A)
and found that 99.83% of pixels have a CH$_4$ emission value of <100 TgCH$_4$yr$^{-1}$.
Therefore, the 4000 storage tank pixels and 4000 background object pixels are
randomly selected from pixels with a CH$_4$ emission value of <100 TgCH$_4$yr$^{-1}$. As shown
in Figure 10B, the CH$_4$ emission values of storage tank pixels are statistically
significantly larger than that of background object pixels with a p-value <0.05. It
indicates storage tanks are significant energetic sources of $CH_4$ emission. With our
proposed dataset STD, it is possible to monitor the greenhouse gas emissions from
storage tanks to take effective measurements for potential climate warming reduction
in time.

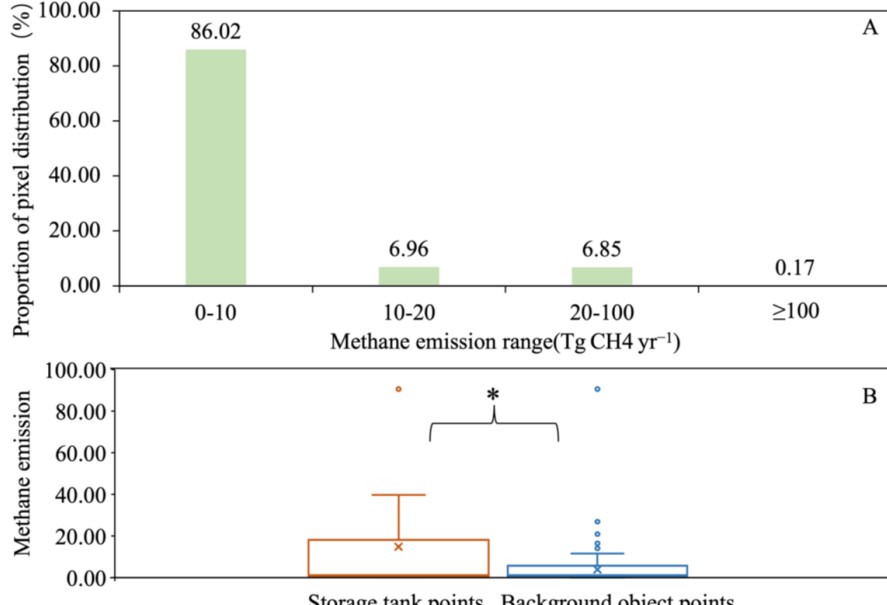

*Figure 10. Distribution pattern of storage tank pixels with different $CH_4$ emission*
*estimations: (A) Proportion of pixels with different $CH_4$ emission estimations; (B) box*
*plot of $CH_4$ emission of storage tank points and background object points.*

**5.3 Temporal impact on $CH_4$ emission**
Given the constraints of historical high-resolution imagery on Google Earth, the
earliest ascertainable construction year for storage tanks is set to 2000, with the latest
capped at 2021, as depicted in Figure 11. Therefore, our dataset STD includes storage
tanks constructed in years of 2000-2021. It is noted that storage tanks were largely
constructed in 2009, 2010, 2012, 2013, and 2014, while those in 2000 and 2001 were
less constructed, with quantities of approximately twenty. To align with the construction
temporal range of storage tanks in the dataset, $CH_4$ emission products of 2005, 2010,
2015, and 2020 are utilized, as these emission products are updated every five years.
To explore the impact of storage tank construction on $CH_4$ emission, the storage tanks
are grouped by the product year of $CH_4$, as listed in Table II. Storage tanks built in the
years 2000 and 2021 are excluded from the impact analysis due to the exceed of the
corresponding impact temporal range of $CH_4$ emission.
Table II. Correspondence between the year of $CH_4$ emission product and group of
construction years of storage tanks.

| Year of $CH_4$ emission product | Year group of storage tanks constructed |
|---|---|
| 2005 | 2001-2005 |



| 2010 | 2006-2010 |
|---|---|
| 2015 | 2011-2015 |
| 2020 | 2016-2020 |


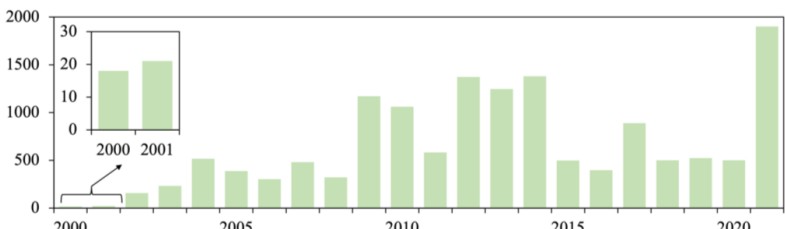


*Figure 11. Number of storage tanks constructed in different years.*

It is noted that the spatial resolution of the $CH_4$ emission product is coarser than
the images we used to generate our proposed STD dataset; similar to the works in spatial
consistency exploration, the storage tanks constructed in different groups of years are
gridded by the $CH_4$ emission product, and the density of storage tanks is calculated for
each grid. We conducted a correlation analysis to explore the statistical significance of
the impact of storage tank construction on $CH_4$ emission over 2005-2020 at levels of
$p=0.05$ and $p=0.1$, respectively. Moreover, the rate of $CH_4$ emission change and oil tank
density newly constructed every five years are calculated according to Equation (8) and
demonstrated accordingly in Figure 12.
$$R=(I_{2020}-I_{2005})/4 \tag{8}$$

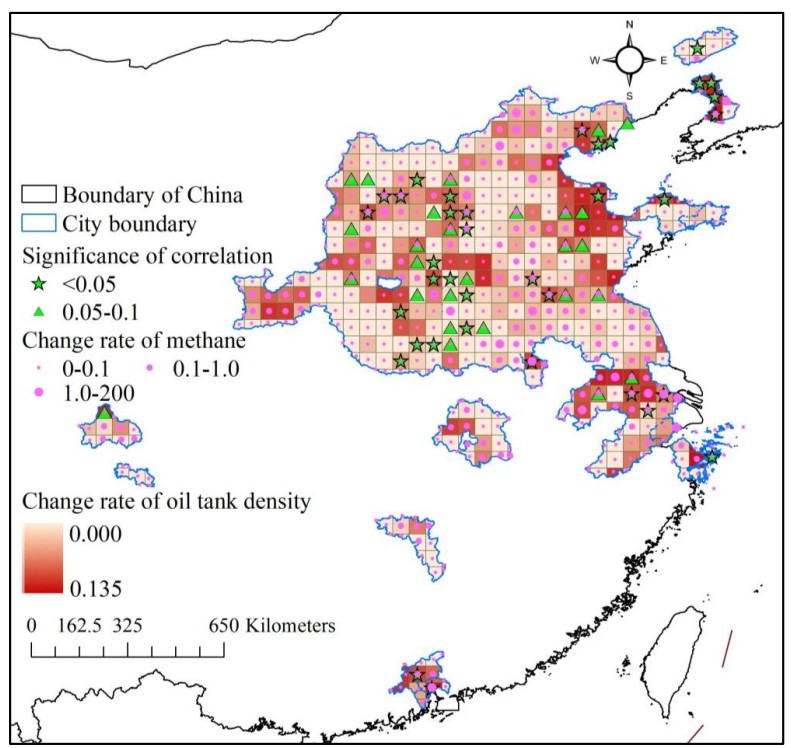

*Figure 12. Significance of correlation between change rate of oil tank density and*
*CH₄ emission change.*

Both CH$_4$ emission and newly constructed storage tank density increased from
2005 to 2020, with positive rates in Figure 12. Over the 92 cities in this study, storage
tanks are constantly being constructed to meet the industrial demand, but CH$_4$ emission
is continuously increasing too. The storage tanks of cities such as Yingkou, Panjin,
Dongying, Binzhou, Yantai, Weifang, Tangshan, Linyi, Rizhao, Puyang, Xi'an,
Pingdingshan, Huainan, Nanjing, Maanshan, Changzhou, Wuxi, Chengdu, Foshan,
Dongguan, and Guangzhou are constructed with higher rates than the other cities. CH$_4$
from energetic activities is emitted at a highly increasing rate in multiple cities, such as
Beijing, Yingkou, Zhenjiang, Nanjing, Maanshan, Changzhou, Wuxi, Shijiazhuang,
Huainan, and Dongguan. Grids showing a statistically significant correlation ($p<0.1$)
between storage tank density and CH$_4$ emissions typically display a notable rise in the
rate of storage tank density, particularly in grids with a p-value less than 0.05. This trend
suggests that areas with active storage tank construction may contribute significantly to
increased CH$_4$ emissions. Some grids exhibit high CH$_4$ emission increasing rates but
low storage tank density increasing rates. This pattern suggests that while storage tank
construction significantly contributes to CH$_4$ emissions, other sources related to energy
production, such as the extraction and transport of coal, oil, and natural gas, are also
major contributors to CH$_4$ release. However, regarding the 92 typical cities with
intensive storage tank distribution and construction, the impact of storage tank





construction on CH$_4$ emission from energetic activities is largely statistically significant,
especially in areas with a high rate of new storage tank construction. Therefore, it is
necessary to propose effective measurements to mitigate CH$_4$ emissions from the
continuously constructed storage tanks.

**6. Discussion**
**6.1 Comparison with published Datasets**
To the best of our knowledge, limited research has been published concerning
remote sensing datasets on storage tanks. The dataset, NEPU–OWOD V1.0, is a
recently proposed oil storage tank dataset featuring 1,192 oil storage tanks from 432
images of Google Earth. It covers the city of Daqing on a limited scale. However, the
dataset lacks georeferenced information, hence the difficulty in supporting further
research by governmental agencies and academic groups on various subjects such as
air pollution control and energy consumption balance studies (Wang et al., 2021). This
is similar to the NEPU–OWOD V1.0 dataset, the Oil and Gas Tank Dataset (Rabbi et
al., 2020), which comprises 760 image patches of size 512×512. The images are taken
at a spatial resolution of 30 cm, and the annotations are boundary boxes rather than
details on the exact shape. To assess the national energy demand, an oil storage tank
dataset is released on the platform Kaggle (Heyer, 2019). However, the images are
collected from Google Earth without georeferenced information. Only 100 image
patches of size 512×512 pixels are included in the dataset. Publication of datasets on
oil storage tanks is generally developed to improve automatic methods for the detection
of storage tanks rather than further environmental analysis based on the combination
and synthesis with datasets of other domains, such as air pollution products. Therefore,
the proposed STD dataset is the first storage tank inventory that provides a detailed
distribution of storage tanks of diverse sizes in 92 cities in China. Each storage tank in
the dataset has undergone rigorous verification by six experts. Additionally, the dataset
meticulously logs the construction year for each tank. This allows for an analysis of the
temporal evolution of storage tank distribution and its combined effects with CH$_4$
emissions on the climate. Such insights pave the way for developing more effective
energy management and climate change mitigation strategies, serving as a valuable
resource for research in atmospheric science, environmental studies, and sustainable
development.

**6.2 Uncertainties, limitations, and implications**
The Storage Tank Dataset (STD) we've compiled for 92 cities in China serves as
a valuable tool for climate change research, despite certain limitations. The extraction
process from high-resolution images is subject to inaccuracies due to shadows and the
inherent limitations of representing three-dimensional tanks as two-dimensional circles,
potentially leading to slight positional errors (Figure 13A). Additionally, the variance
in perspective between our collected high spatial resolution images and Google Earth
historical images can cause deviations in visual refinement in the tanks' vectorized
outlines (as shown in Figure 13B). To mitigate these issues, expert analysis is employed





to ensure tank identification and location precision, referring to the collected high
spatial resolution images.

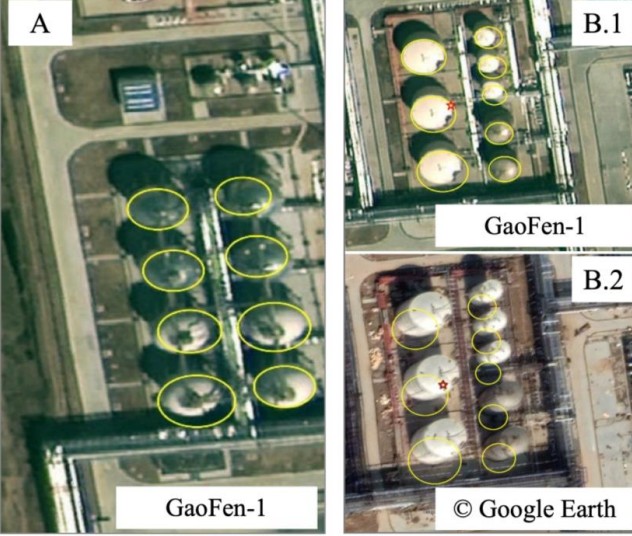

*Figure 13. Example cases of our proposed STD dataset: (A) cases with shifted circles*
*due to cast by shadow; (B) cases with largely deviated circles in different images due*
*to different viewing angles.*
The pioneering STD dataset encompasses georeferenced storage tank shapes for
92 key Chinese cities crafted from high-resolution images. For each storage tank, the
corresponding construction year is assigned, referring to the high-resolution historical
images of Google Earth. It's a versatile resource with spatial and temporal distribution
patterns, not just for mapping $CH_4$ and other emissions but also for aiding the
development of infrastructural strategies across various industries. However, the dataset
currently lacks volumetric data due to the absence of height measurements for the tanks.
Future enhancements aim to incorporate height data through advanced remote sensing
technologies like SAR imagery, enriching the dataset with three-dimensional accuracy
and providing a more comprehensive understanding of storage tank capacities.

**7. Dataset availability**
The STD dataset is publicly available as a repository at
https://zenodo.org/records/10514151 (Chen et al., 2024). The dataset is provided in a
shapefile, wherein a polygon with an area attribute in units of m2 represents each
storage tank, and two attributes of years, year_1 indicating the most recent year when
a storage tank was absent (last year image without storage tank) and year_2 indicating
the earliest year when it was first observed (first year image with storage tank). The
inventory is intended to be used to further analyze the impact on CH4 emissions, devise
and implement more efficient energy management strategies. Moreover, our approach
represents a powerful new source to improve automatic methods for storage tank
extraction from high spatial resolution images, given that it represents a comprehensive
and state-of-the-art inventory with tens of thousands of storage tanks georeferenced of
92 typical cities over China.

## 8. Conclusions


In support of $CH_4$ emission control to mitigate climate warming, the STD dataset
is proposed by providing a meticulously georeferenced inventory of storage tanks larger
than 500 $m^2$ across 92 key cities of China in years of 2000-2021. Leveraging a novel
semantic segmentation framework, Res2-UnetA, and rigorous visual interpretation
based on the collected high spatial resolution images, historical high spatial resolution
images from Google Earth, and field survey, the dataset not only details the spatial
distribution of large storage tanks but also includes their construction years. Based on
the STD dataset, the spatial distribution pattern of the storage tanks of different sizes
was analyzed in 92 cities. We also explored the impact of storage tank construction on
$CH_4$ emission from energetic activities through 2005-2020. Compared with the
published datasets for storage tanks, the STD dataset is the first inventory that compiles
georeferenced storage tanks in 92 cities with detailed shape boundaries and construction
years. In general, publicly available datasets on storage tanks typically cover only part
of a city without georeferenced information and detailed shape boundaries. It is,
therefore, difficult to objectively explore the extent and patterns of environmental
impact and the energy management of the storage tanks at large scale. The STD dataset
enables large-scale environmental impact analysis of storage tanks and their correlation
with $CH_4$ emissions. It demonstrates strong spatial consistency with $CH_4$ emissions in
92 typical Chinese cities, highlighting the substantial increase in $CH_4$ emissions due to
storage tank construction. The storage tank dataset STD can contribute significantly to
supporting energy management strategies and sustainability development studies while
giving direct support to academic research and government agencies.

## Author contributions


FC and LW designed the study and conducted the experiments. YW, HZ, NW, PM and
BY compiled the dataset. BY wrote the manuscript.

## Competing interests


The authors declare that they have no conflicts of interest.

## Financial support


This work was supported by the National Key R&D Program of China (No.
2022YFC3800701), the Youth Innovation Promotion Association, CAS (2022122), the





China-ASEAN Big Earth Data Platform and Applications (CADA, guikeAA20302022).

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
