# Peer review of "Retrieval of dominant methane (CH4) emission sources, the first high resolution"

_Earth System Science Data, 2024_

## Author Response (AR1)

Dear editor,

Thanks for your generous help with our paper and the reviewers' valuable comments concerning our manuscript entitled "Retrieval of dominant methane ($CH_4$) emission sources, the first high resolution (1-2m) dataset of storage tanks of China in 2000-2021" (essd-2024-10). These comments are very helpful in revising and improving our paper, as well as guiding our future research. We have studied the comments carefully and tried our best to revise the manuscript. Detailed response to each comment is listed as follows:

**Reviewer #1:**

This manuscript generates a storage tank dataset (STD) by implementing a deep learning model with manual refinement based on high spatial resolution images (1-2m) from the GaoFen-1, GaoFen-2, GaoFen-6, and Ziyuan-3 satellites over some region in China in 2021. Since oil gas infrastructure data is barely publicly available in China, this dataset could give researchers a tool to estimate or scale up methane emissions from relevant industry sectors, and thus it is of value to be published in this journal.

The reviewer has several comments which may help improving the quality of the manuscript:

**Q1. Targeted tanks: The manuscripts screens tanks under two criteria: footprint bigger than 500 m2; located within the built area and bare land in the LULC product of the Esri Land Cover in 2021.While the objective of this study is to retrieve dominant methane emission sources, the reviewer is wondering how many of tanks retrieved are methane emission sources, and how many tanks which are methane emissions sources, are neglected. For example, oil gas batteries are major methane emission sources, and located in the rural agriculture area or desert. Will these batteries be identified under the current algorithm. What is the rational for 500 m2 threshold? Are there tanks who have footprints smaller than 500 m2?**

**To make this clear, the reviewer would suggest a table summarizing the key features of tanks, including the following aspects: Usage/function; typical size (water volume, m3); typical footprint (m2); typical location (urban, rural?); industry sector.**

**Answer:** Thanks very much for your suggestion. The storage tanks are extracted from remote sensing images, according to their spectral, textual, and morphological features based on deep

learning frameworks as illustrated in our manuscript. It is very difficult to tell the exact usage of each storage tank only according to the appearance in the image, since storage tanks may take similar appearances with different functions. Moreover, it is very difficult to estimate typical size of each storage tank because of lack of height. The typical footprint in the unit of $m^2$ has been provided in the attribute table of our dataset. The storage tanks located in rural agriculture area or desert are not considered in our proposed dataset, because storage tanks of crude oil or other petroleum, and industrial materials, such as alcohols, gases, or liquids, are among the most significant sources of emitting $CH_4$ (Im et al., 2022; Johnson et al., 2022), and they are mostly located in urban area or development zones in China due to the transportation of pipeline. Therefore, we did not distinguish the typical location of the storage tanks in our dataset since they are all located in urban area.

In terms of the threshold of 500 $m^2$, it is determined based on the spatial resolution of remote sensing images we collected to construct the dataset. The storage tanks with 125 pixels or more in remote sensing images can be distinguished more accurately from the complicated background objects. There are tanks with footprints smaller than 500$m^2$, but they are more difficult to be extracted accurately due to the small size. Moreover, since large capacity storage tanks are known to release significant levels of $CH_4$, our proposed inventory focuses on storage tanks with an area of no less than 500 $m^2$. The corresponding sentence has been revised to be 'Given that large capacity storage tanks are known to release significant levels of $CH_4$, resulting in climate warming, the proposed inventory focuses on storage tanks with an area of no less than 500 $m^2$' in **Line 405-407** of the revised manuscript.

**Q2. Dataset accuracy verification. The reviewer did not see any verification of the dataset in the manuscript. Have the researchers verified their findings, for example, checking the accuracy of the dataset for one city/region? How many true positive, false positive? How many tanks are missed? Without such verification process, this is not a complete stand-alone paper.**

**Answer:** Actually, each storage tank of our proposed dataset has been manually validated and refined by six experienced experts through visual interpretation based on our collected high spatial resolution images and field survey to facilitate the corresponding construction year. Therefore, we did not conduct dataset evaluation specifically because each storage tank has been verified and refined. To make the manuscript easier to understand, the corresponding sentence has been modified

to be 'To facilitate the dating of each storage tank's construction year, the reconstructed circle for each extracted storage tank has been manually validated and refined by six experienced experts through visual interpretation based on our collected high spatial resolution images and field survey.' in **Line 371-374** of the revised manuscript.

**Q3. Construction year assignment. The researchers identified the construction year of tanks by the high-resolution historical images available on Google Earth. Can the researchers also identify how many tanks disappear during the period of study?**

**Answer:** The storage tanks extracted in this study are those existing in the high spatial resolution images we collected in year of 2021, and the corresponding construction year is determined for each extracted storage tank referring to the historical images available on Google Earth. Due to the large scale of our study area and the limited availability of high spatial resolution images, the storage tanks are extracted based on mono-temporal images, rather than time-series images. Therefore, it is difficult to identify the disappeared storage tanks during the period of study.

**Minor Observations:**

**Q1. Line 53, " … with 85 times more climate warming potency than CO2": please specify the time length of 85 times.**

**Answer:** Thanks very much for your suggestion. We have specified the time length in **Line 52-54** of the revised manuscript as 'Meanwhile, $CH_4$ is more effective in trapping heat, with 85 times more climate warming potency than CO2 for a decade or two (Stocker, 2014).'.

**Q2. Line 108, "size > 500 m2". Tank size is usually characterized by its water volume. Suggest changing tank size to tank footprint.**

**Answer:** Thanks very much for your generous suggestion. We have changed tank size to tank footprint throughout the revised manuscript.

**Q3. Line 114. "cities": city in English usually refers to urban area. Here it looks like that the city covers both the urban area, as well as the rural counties. Would suggest to change the "city" to "region", and explain the true meaning of it.**

**Answer:** Your suggestion is much appreciated. The word region is better, but it is difficult to describe the administrative boundary of our study area. Therefore, we have modified city to city region throughout the revised manuscript. Hopefully, it is clearer this time.

**Q4. Line 168, digital elevation model (DEM), and Figure 1: Is DEM a model or elevation? In figure 1. DEM High 4708, Low -91. What's the unit for DEM? Please clarify, and add unit in Figure 1 (m?)**

**Answer:** The unit for DEM is meter. Thanks very much for your suggestion. We have added the unit in the caption of Figure 1 as 'Figure 1. Study area demonstration with digital elevation (in the unit of meter) from the Shuttle Radar Topography Mission (SRTM) product.' in **Line 178-179** of the revised manuscript.

**Q5. Line 204 "Given that storage tanks are constructed mainly in residential areas": Please elaborate why. To the reviewer's knowledge, most oil gas storage tanks should be outside of the residential areas.**

**Answer:** The storage tanks are mostly constructed in urban area because of the high expense of pipeline transportation. Moreover, according to our field survey, the storage tanks, especially storing oil and gas, are mostly located in industrial area of urban area. To make the manuscript easier to understand, the corresponding sentence has been modified to be 'Given that storage tanks are constructed mainly in urban area due to the high expense of transportation of pipelines' in **Line 205-206** of the revised manuscript.

**Reviewer #2**

The author created a dataset of storage tanks using a deep learning approach applied to high-resolution remote sensing imagery. This dataset provides a comprehensive, validated, and geo-referenced collection of details, including the precise locations, distributions, and construction years of storage tanks. It covers 92 representative cities, encompassing a total of 14,461 tanks. The manuscript also explores the spatial correlations between the distribution and density of storage tank and methane emissions, contributing to a more profound understanding of the societal, ecological, and settlement impacts of methane emissions from these structures. The paper's innovation lies in

its comprehensive database of storage tanks across 92 typical cities and its large-scale exploration of the spatial interplay between storage tanks and methane emissions. However, there are some problems that require responses from the authors. Afterwards, this manuscript could be accepted for publication after a minor revision.

Some detailed problems:

**Q1. Line 167-168: The author seems to refer to 'coastal cities' when mentioning that "Many of the cities are located near or next to the boundary of 167 mainland China.", and this should be explicitly identified.**

**Answer:** Thanks very much for your comment. The corresponding sentence has been modified to be 'Many of the city regions are coastal cities' in **Line 168** of the revised manuscript.

**Q2. Line 172-173: The manuscript predominantly focuses on the measurement of methane emissions, rather than the measurement of methane reduction.**

**Answer:** Your suggestion is much appreciated. We have modified methane reduction to methane emissions in the revised manuscript in **Line 173-174** as follows:' The lack of efficient measurements in $CH_4$ emissions will result in a more direct impact on the populations in the residential area'

**Q3. Line 204-205: Why are the storage tanks mainly constructed in residential areas? I think it should be placed in the built area and bare ground that is far from residential areas in urban settings.**

**Answer:** Sorry for the misleading sentences. We have modified the corresponding sentence to be 'Given that storage tanks are constructed mainly in urban area due to the high expense of transportation of pipelines' in **Line 205-206** of the revised manuscript.

**Q4. Line 220-221: The terms of LULC categories should be consistent throughout the manuscript. For instance, the "Bare ground" or "Bare Ground" and "Flooded vegetation" or "Flooded Vegetation" should be maintained in a uniform format.**

**Answer:** Sorry for the inconsistent expressions. We have modified the inconsistent' Bare Ground' and 'Bare ground' to be 'bare ground'; 'Flooded Vegetation' and 'Flooded vegetation' to be 'flooded

vegetation' in the revised manuscript to keep consistency.

**Q5. Line337-343 & 508: The equations should be aligned at the centre within the text.**

**Answer:** Thanks very much for your suggestion. All the equations have been revised to place in the center in the revised manuscript.

**Q6. Line 425-426 & 598 & 602: The "m2" should be written as "m2" and the "CH4" should be written as "CH4" in the text.**

**Answer:** Sorry for the wrong writing. We have modified 'm2' to be 'm$^2$', and 'CH4' to be 'CH$_4$' in the revised manuscript.

**Q7. Line 428-430: The storage tanks of different categories should not omit their respective units in the figure, such as 500-1000 m2.**

**Answer:** Sorry for the omission. We have added unit for all the footprint size of storage tanks throughout the revised manuscript.

**Q8. Line 448: The author seems to refer to 'coastal regions' when mentioning "especially at the border of mainland China", and this should be explicitly identified.**

**Answer:** Thanks very much for your pointing this out. We have modified the sentence to be 'There are also some city regions with a high density of storage tanks and low CH$_4$ emission estimation, especially coastal cities, as in the cases of F.' in **Line 457-458** of the revised manuscript.

**Q9. Line 466 & 468: The units "TgCH4yr-1" should be separated in the text.**

**Answer:** Yes, you are right. We have separated the unit 'TgCH4yr-1' to be 'Tg CH$_4$yr$^{-1}$' in the revised manuscript.

**Q10. Line 470 & 525: The p-value is usually represented as "p=0.1" or "p=0.05".**

**Answer:** Thanks for pointing this. We have modified the corresponding sentence to be 'As shown in Figure 10B, the CH$_4$ emission values of storage tank pixels are statistically significantly larger than that of background object pixels at a confidence level of p=0.05' in **Line 478-480** and 'Grids

showing a statistically significant correlation (p<0.1) between storage tank density and CH$_4$ emissions typically display a notable rise in the rate of storage tank density, particularly in grids with at a confidence level of p=0.05' in **Line 534-537** of the revised manuscript.

**Q11. Line 475-476: In figure 10 (A), the "CH4" should be "CH4". In figure 10 (B), the unit "Tg CH4 yr-1" of methane emissions of left panel is necessary after "Methane emission".**

**Answer:** Sorry for the error we made. 'CH4' has been corrected to be 'CH$_4$' in the revised Figure 10(A). The unit 'Tg CH$_4$ yr$^{-1}$' has been added in the caption of Figure 10(B). For your convenience to check, the revised Figure 10 has been listed below.

[Figure]

*Figure 10. Distribution pattern of storage tank pixels with different CH$_4$ emission estimations: (A) Proportion of pixels with different CH$_4$ emission estimations; (B) box plot of CH$_4$ emission (Tg CH$_4$ yr$^{-1}$) of storage tank points and background object points.*

**Q12. Line 490 & 493: The formatting of tables should be consistent throughout the entire manuscript, using either "Table 1, Table 2" or "Table I, Table II".**

**Answer:** Sorry for the inconsistent error. We have modified Table 1 to be Table I in the revised manuscript.

**Q13. Line 505: The term 'oil tank' is being introduced here for the first time. It is necessary to clarify whether 'oil tank' and 'storage tank' are synonymous with each other.**

**Answer:** Sorry for the misleading expression. We have modified 'oil tank' to be 'storage tank' in the revised manuscript.

**Q14. Line 676 & 715 & 724 & 725 & 734 & 735 & 743 & 749 & 778: The author should undertake a thorough review to guarantee the entirety of these references.**

**Answer:** Sorry for the missing information of the references. We have checked the references and added the corresponding details in the corresponding sections of references.

**Reviewer #3**

The authors used their own deep learning model, Res2-Unet+, to generate the first high-resolution 1 (1-2m) dataset of storage tanks from over 4000 images and assign the year of each storage tank. The dataset is then used for CH4 emission analysis. The dataset is useful and the analysis is reasonable. It is suitable for publication in ESSD. However, I have the following comments:

**Q1. Abstract Line 30 "based on high spatial resolution images (1-2m) …". I suggest the authors add the number of images.**

**"based on over 4000 high spatial resolution images (1-2m)… "**

**Answer:** Thanks very much for your suggestion. We have revised the corresponding sentence to be 'we generated a storage tank dataset (STD) by implementing a deep learning model with manual refinement based on 4,403 high spatial resolution images (1-2m)' in **Line 28-30** of the revised manuscript.

**Q2. 3.3 Land use land cover product. Line 263-264. "historical high spatial resolution images, high spatial resolution images collected, and field survey from Google Earth", please refine the sentence. from my understanding, only historical high spatial resolution images are from Google Earth.**

**Answer:** Thank you for your suggestion. We have revised the sentence correspondingly.

**Q3. Section 4.2.1**

**Line 309 "Res2-Unet+ by Yu et al. (Yu et al., 2021)". "by Yu et al." should be removed. The**

**same issue is in line 352.**

**Line 329 "Our proposed Res2-UnetA", does it mean Res2-Unet+?**

**equation (1)-(7), what do "f, m, n, h, etc" stand for?**

**Answer:** Thanks very much for your suggestions. We have removed in **Line 312** of the revised manuscript. Res2-UnetA is newly proposed in this work, stemming from Res2-Unet+, as illustrated in **Line 311-313**.

In terms of Equations (1)-(7), $f$ indicates feature map, $m$ and $n$ are the size of feature map $f$, $h$ is the channel number of feature map $f$. Sorry for the missing introduction. We have revised the corresponding sentences in **Line 322-323** and **Line 337-348** as follows:

'Detailed calculation of channel-wise and spatial attention modules can be referred to Equations (1)-(7).

Spatial average pooling (sa) and spatial maximum pooling (sm) operations are calculated as the average value and maximum value of input feature map $f$ with size of $m \times n$, as described in Equations (1)-(2). Correspondingly, the channel-wise average (ca) and maximum pooling (cm) operations are the average feature values of all the $h$ channels and the maximum feature values of all the channels in Equations (3)-(4). The output feature map of the spatial attention module (SA) and channel attention module (CA) are calculated according to Equations (5)-(6), respectively, and the synthesis of the feature maps from the channel and spatial attention modules is organized by multiplication, as illustrated in Equation (7).'

**Q4. Section 4.3 The STD dataset covers 2000-2021. How do you define the year if the storage tank was built before 2000, and how many such cases are there?**

**Answer:** Due to the limited accessibility of high spatial resolution images before 2000 from Google Earth, the storage tanks with the record of first year image with storage tank, but without that of last year image without storage tank in our proposed dataset STD are possibly constructed before the year of 2000. Upon checking the storage tanks in our dataset, there are 188 storage tanks lacking the record of last year image without storage tanks due to lacking of continuous historical high spatial resolution images from Google Earth, which are possibly constructed in year before 2000.

To make the manuscript easier to understand, we have added the corresponding sentence in **Line 396-399** as 'For the storage tanks built before 2000, they are recorded with the first year image with

storage tank in the shapefile, but lacking the last year image without storage tank in our proposed dataset STD due to the limited accessibility of high spatial resolution images before 2000 from Google Earth'

**Q5. Section 5.1 Line 404-405. "It may be seen that 404 storage tanks of 500-1000 m2 are more than those of larger sizes". Then, what are the criteria for threshold 500m$^2$?**

**Answer:** The threshold 500m$^2$ was selected by experience, given that large storage tanks may emit significant levels of $CH_4$.

**Q6. Section 5.2 What is the year for density calculation in Fig 9?**

**Answer:** The storage density calculation in response to different $CH_4$ emissions in the atmosphere is explored in year of 2020. We have added the corresponding illustration in **Line 444-445** as 'we explored the spatial consistency between estimated $CH_4$ from energy emission products in year of 2020 and the density of storage tanks in our proposed dataset STD over the study area.'

**Q7. Section 6.1**

**NEPU-OWOD dataset, an oil storage tank dataset from platform Kaggle, and the STD dataset. They should be separated into different paragraphs.**

**Answer:** Thanks very much for your suggestion. We have separated the illustration of our proposed STD dataset over the published works of NEPU-OWOD V1.0 dataset, the Oil and Gas Tank Dataset, and the oil storage tank dataset from platform Kaggle into two paragraphs in the revised manuscript.

**Q8. References. The references need to revised in uniform format. Some references are missing. Some references have doi.**

**Answer:** Sorry for the uniform format of references. We have checked and revised all the references to maintain consistency.

We are deeply appreciated for your kind help and the reviewers' careful work honestly, and hope that our modifications will meet the approval. Please let me know if this paper needs any further

modification.

We look forward to your information about our revised paper.

Best regards,

Fang Chen